# Daily Variability in Sedentary Behaviour and Physical Activity Responsiveness in Older Women

**DOI:** 10.3390/s25072194

**Published:** 2025-03-30

**Authors:** Dale M. Grant, David J. Tomlinson, Kostas Tsintzas, Gladys L. Onambele-Pearson

**Affiliations:** 1Health & Exercise Medicine, Division of Public Health, Sport and Wellbeing, School of Allied and Public Health, Faculty of Medicine, Health and Society, Exton Park Campus, University of Chester, Parkgate Rd, Chester CH1 4BJ, UK; dale.grant@chester.ac.uk; 2Department of Sport & Exercise Sciences, Institute of Sport, Manchester Metropolitan University, Manchester M15 6BX, UK; david.tomlinson@madeley.set.org; 3Madeley Hight School, Newcastle Road, Madeley, Crewe CW3 9JJ, UK; 4Queen’s Medical Center, School of Life Sciences, University of Nottingham Medical School, Nottingham NG7 2UH, UK; kostas.tsintzas@nottingham.ac.uk

**Keywords:** inter-day variability, LIPA, physical activity, regularity, sedentary

## Abstract

Free-living physical behaviour (PB), from sedentarism through to vigorous physical activity (PA), is increasingly studied due to its links to health outcomes. However, it remains unclear whether pre-existing day-to-day regularity in certain PB patterns influences intervention responsiveness. Therefore, this study hypothesized that (1) inter-day variability in certain PBs would decrease following a sedentary behaviour (SB) reduction intervention, and (2) those with high inter-day variability (low regularity) at baseline would be less likely to alter their behaviour compared to those with low inter-day variability (high regularity). Thirty-six older women (73 ± 5 years) were allocated to one of three groups: (1) daily SB fragmentation (SBF) (n = 14), (2) single daily bout of continuous light-intensity PA/LIPA (n = 14), or (3) control (n = 8), where no instructions vis-à-vis altering daily physical activity or sedentary behaviour were given. PB was objectively assessed (weeks 0 and 8) using three-dimensional accelerometry. Participants (48% of the study sample) with high regularity at baseline (<25th sample percentile for SB and PA bout length), showed greater SB reduction, and increased average PA bout length (*p* < 0.05) at week 8. These findings suggest that baseline regularity in physical behaviour may enhance intervention responsiveness. This aligns with theories of habit formation and self-regulation, indicating that personalised interventions would benefit a wider range of populations.

## 1. Introduction

Daily physical behaviour (PB), or physical mobility actions during waking hours, includes—at the lower end of its spectrum—sedentary behaviour (SB) or low levels of energy expenditure (≤1.5 metabolic equivalents) and a seated or reclined posture [1,2]. SB is associated with obesity [3], cardiovascular disease [4], all-cause mortality [5], and frailty [6], but higher cognitive function [7,8]. Importantly, the negative health impact of SB is present even after accounting for any co-existing moderate-to-vigorous physical activity (MVPA) [5,9,10,11]. This may be due to the natural displacement of beneficial standing time [12] and light-intensity PA (1.5–3.0 metabolic equivalents) time [13,14,15,16], as well as the negative effects of prolonged sedentary behaviours (i.e., longer SB bouts) [17,18,19].

The evidence base has largely been through objective PB measures (e.g., tri-axial accelerometry data) over a number of days [11]. Such studies typically investigate whether an association exists between a specific health outcome (e.g., all-cause mortality during follow-up) and the average monitoring period value for a PA- or SB-based outcome [11]. These PA and SB outcomes are usually presented as either a weekly or a total accumulated amount of time. In contrast, the PA/SB or PB accumulation pattern (e.g., average SB bout length over the course of 7 days) expresses the relative amount of PA over SB. The PB accumulation pattern has also been investigated with regard to its association with health outcomes [18].

Whilst average values for absolute PB and PB accumulation patterns are a useful means of statistical reporting, they may inadvertently hide important information. One such overlooked metric is inter-day variability. Thus, a higher intra-week variability (i.e., co-efficient of variation) would represent a PB that is performed in a sporadic manner day-to-day [20]. In contrast, a lower intra-week variability would represent a behaviour that is performed in a more consistent day-to-day fashion.

Many PA recommendations already place particular emphasis on regular PA, e.g., ‘do 30 min on at least 5 days a week’ [21]. Interestingly, the 2019 UK chief medical officer’s guidelines removed this ‘5 days per week’ recommendation, instead emphasizing ‘150 min can in fact be accumulated in bouts of any length’ [22]. The meta-analytic review cited in support of cumulative activity states “adults are likely to accrue similar health benefits from exercising in a single bout or accumulating activity from shorter bouts throughout the day” [23]. As such, evidence supports the fact that PA of any bout duration (e.g., including sporadic bouts of MVPA ≤ 10 min in length) improves health [24], regardless of the intra-day pattern. However, it is unclear what effect, if any, inter-day pattern of PA may have on health.

Inter-day variability is a recognized method of monitoring athletes’ training load [25,26,27]. However, there is no existing standard for measuring variability in physical behaviour in healthy populations. Previous studies have examined the variability in physical behaviour in clinical populations, by examining the influence of week vs. weekend day [28,29]. Interestingly, a previous study, found that adults with the greatest regularity of daily steps (inter-day coefficient of variation ≤ 40%) reduced their weight by 0.33 kg more compared to those with the lowest daily steps regularity (CV ≥ 63%) [20], which suggests that PA variability may be a biomarker for the impact of PA on health [20].

Interestingly, whilst inter-day light-intensity physical activity (LIPA) variance (assessed by heart rate, for example) appears to be lower in older vs. younger populations [30], potentially owing to an adult’s daily routine generally following more of a consistent routine [31] or adults generally having a lower PA amount [32]. MVPA, on the other hand, displays a higher inter-day variability in the elderly compared to younger adults—though not in children—irrespective of gender [33,34]. On the other hand, SB exhibits a lower inter-day variability compared to more active behaviours in children and younger adults [33,34]. This is also likely to be the case in an adult population, considering that SB is firmly embedded into people’s habitual routines (especially older adults) [35] and that people are largely unaware of how much inactivity they perform [36]. Older adults are particularly relevant to sedentary behaviour interventions, as they tend to have more structured daily routines and lower overall physical activity levels [35], which may influence their ability to modify behaviour patterns. Additionally, given the well-documented associations between excessive sedentary time and adverse health outcomes in ageing populations [3], understanding how inter-day variability impacts intervention responsiveness in this group is critical for developing effective strategies to promote healthier movement behaviours. However, to the current authors’ knowledge, PA and SB inter-day variability have yet to be investigated in the context of older adult populations, different PA/SB metrics, and interaction with behavioural change adherence. Considering women generally engage in less PA compared to men [37], there is also a need to investigate inter-day variability in a gender-specific manner.

Therefore, the aims of this study were focused to (1) determine whether PA/SB inter-day variability changes following SB reduction/fragmentation interventions in older women and (2) examine whether baseline PA/SB inter-day variability impacts a participant’s ability to change their PB in response to an SB reduction/fragmentation intervention. It was hypothesized that (1) the inter-day variability in certain physical behaviours would reduce (i.e., increased regularity) in response to the SB reduction intervention, and (2) those older women classified as having a high inter-day variability (i.e., low regularity) at baseline would be less likely to alter their behaviour compared to those classified as low inter-day variable at baseline (i.e., high regularity).

## 2. Materials and Methods

### 2.1. Experimental Design

All data were collected between June 2018 and July 2019 from a group of community-dwelling older women. The study was approved by the ethical committee of Manchester Metropolitan University in March 2018 [approval code: 230118-ESS-DG-(2)]. Recruitment packages (which included health questionnaires, participant information sheets, and a pre-paid return envelope) were sent to all contacts (n = 271) on a pre-existing research database of older women. Inclusion criteria were being female, aged between 65–85 y, and not knowingly allergic to the physical behaviour monitoring equipment. Exclusion criteria included currently suffering from disease (active cancer, uncontrolled diabetes, and cardiovascular disease), recent history (<6 months) of lower limb muscle/tendon/joint disorder (e.g., osteoarthritis), or currently using prescribed medications likely to affect movement (e.g., dopamine antagonists). An a priori power analysis was not performed due to a lack of previous investigations examining changes in physical behaviour regularity in older adults, highlighting the novelty of the current study. As such, estimation of the required sample size to detect significant changes in the desired outcomes was based on previous SB interventions in older adults that utilized ~25–38 participants [38,39,40]. The current achieved sample size of 31–36 older women falls well within this range. After screening the participants against the inclusion/exclusion criteria, 36 eligible participants visited the laboratory for familiarization. Prior to any procedures taking place, participants gave written informed consent, in line with the declaration of Helsinki [41].

During the familiarization visit, participants filled in various questionnaires (e.g., falls risk assessment tool) and underwent anthropometric assessment (height and weight). Before leaving, participants were provided with monitoring equipment (see physical behaviour assessment). After seven continuous days of physical behaviour monitoring, participants visited the laboratory again to return equipment. Before leaving the laboratory, participants were allocated to one of three groups—(1) sedentary behaviour fragmentation (SBF) (n = 14); (2) single 45–50 min bout continuous light activity (LIPA) (n = 14); or (3) control, i.e., no lifestyle change (n = 8, a smaller group than initially planned owing to dropouts upon allocation before the study started—such an attrition, whilst unplanned, was expected to have minimal impact on baseline imbalances and study outcomes)—building on previous SB reduction [42] and randomized interventions [38,39,40]. The study design was a randomized controlled trial with three groups (2 experimental groups and 1 control), and 2 time points (before and after the 8-week interventions). Participants received specific instructions based on their allocated group. Participants returned to the laboratory after 8 weeks to return the equipment outlined above. Individual participant compliance was objectively monitored at baseline and at week 8 of the intervention for 4–7 days.

### 2.2. Physical Behaviour Interventions

The purpose of the two intervention groups was to manipulate the protocol for displacing sedentary behaviour time with added daily light-intensity PA (LIPA) (45–50 min in total), as described previously [42], with such LIPA derived from the compendium of physical activities [43]. All groups recorded wake/sleep times within a diary and received fortnightly home visits from the principal investigator (to refresh instructions, replace the movement monitor, etc.).

SBF group: Participants were instructed not to perform sedentary behaviour for more than 30 min at a time, and that for every 30 min of sedentary behaviour performed they should perform 2 min of upright LIPA (e.g., walking, side-to-side shuffling, washing dishes, etc.) SBF participants wore a second accelerometer (activPAL3™ triaxial physical activity logger, PAL technologies Ltd., Glasgow, Scotland, UK) (34 mm × 55 mm × 6 mm) mounted alongside the GENEA unit (see physical behaviour assessment below). The activPAL3™ was pre-configured using activPAL software version 7.0 (PAL technologies Ltd., Glasgow, Scotland, UK) to prompt movement following 30 min of SB, in the form of a vibration against the skin.

LIPA group: Participants were instructed to perform a continuous single bout of 45–50 min LIPA (e.g., walking, side-to-side shuffling, washing dishes etc.), every day for the duration of the 8-week intervention.

Control group: Participants were specifically instructed to maintain their habitual routine.

### 2.3. Physical Behaviour Assessment

Physical behaviour was objectively assessed with a blinded GENEActiv original (GENEA, Activinsights Ltd., Kimbolton, UK) triaxial accelerometer (43.0 mm × 40.0 mm × 16.0 g) that did not provide feedback to the participant. The GENEActiv has previously been validated for objective measures of free-living physical behaviour [44,45,46]. The thigh-mounted position has previously been shown to be both valid and reliable for assessing PB with the GENEActiv device [45,46,47,48].

Once charged, each device was pre-configured to capture and record PB at a frequency of 60 Hertz [samples per second, (Hz)] and set to the maximum monitoring period of 12 days. Following configuration, a 35 mm × 55 mm × 7 mm soft sponge (Vitrex, Burton-upon-Trent, Staffordshire, UK) was attached to the back of the GENEA using two microporous strips (3M, St. Paul, MN, USA). The GENEA was mounted on the participant’s leg (anterior, 50% of femur length) using two Tegaderm Films 1626W (100 mm × 120 mm) (3M, St. Paul, MN, USA). Spare films were provided in case the first strips became loose.

A diary was provided to each participant to record waking/sleep times (time they turned the lights off to go to sleep at night). Such times were used to triangulate sleeping hours. In cases of missing wake/sleep times, the epoched data were manually assessed to determine when daily movement ceased. Participants were asked to record any device disruption (date/time device was removed/re-attached). This allowed such a period to be removed from analysis. Following each monitoring period, the GENEA device was removed from the sponge and thoroughly cleaned. The recorded data from the GENEA device were downloaded onto a computer using the GENEA software (Version 3.2, Activinsights Ltd., Kimbolton, UK). The .bin file was smoothed in the ‘Data Convertor’ Stage into 10 s (s) epochs to optimise the sensitivity to detect changes in physical behaviour [49,50,51].

### 2.4. Physical Behaviour Classification

The Cheshire Algorithm for Sedentarism (CAS) [45,46] was used for offline analysis. The CAS uses cut-off points to identify specific physical activity intensity ranges utilizing metabolic equivalent thresholds (SB, <1.5 metabolic equivalents; LIPA, 1.5–3.0 metabolic equivalents; MVPA, >3.0 metabolic equivalents) mapped against the concurrently recorded GENEActiv gravitational pull and acceleration data [47] with a balanced accuracy for estimating physical behaviour that meets a critical threshold of ≥80% in older adults [47,48].

Considering the fundamental need to determine inter-day variability, the number of valid days within a monitoring period used to determine inter-day variability was a key decision, due to the Hawthorne effect, in combination with practical limitations and considerations such as minimizing participant burden, resource constraints, and study focus. This led us to use a monitoring period of sufficient duration to balance data quality and participant comfort. When wearing an accelerometer, certain participants may react through consciously/unconsciously altering their physical behaviour (termed reactivity) [52]. Thus, a monitoring period of sufficient duration was required to reduce any reactivity effects that may occur. Protocols of 2.5–4.9 valid days of hip accelerometry have been shown to be the minimum number of valid monitoring days required to achieve a good intra-class correlation coefficient (ICC) of 0.80 [53]. Therefore, if a monitoring period for a particular participant (at baseline or week 8) did not contain ≥4 valid days, it was removed from the analysis.

The CAS provided several physical behaviour parameters, which were organized into 4 distinct categories: (1) PA amount [standing (hrs/day), LIPA (hrs/day), moderate-to-vigorous PA (MVPA) (hrs/day), MVPA in bouts of ≥10 min duration (hrs/day), MVPA in bouts of <10 min (mins/day), and daily sum of physical activity bout time (mins/day)]; (2) PA patterns [physical activity bouts (n/day), and average daily physical activity bout length (min/day)]; (3) SB amount [SB (hrs/day), and SB% (%.walkinghrs/day)]; and (4) SB patterns [SB in bouts of <5 min duration (n/day), i.e., short SB counts, average daily sedentary behaviour bout length (mins/day)], etc. [54].

Participants were classified as either being sedentary (≥8 h/day) or ambulatory (<8 h/day) depending on their average daily sedentary behaviour time at baseline. Participants were also further classified as physically active (≥150 min/week MVPA ≥ 10 min bouts), or non-physically active (<150 min/week MVPA ≥ 10 min bouts). Such limits were selected as classification thresholds given that sedentary time appears to be exponentially hazardous above 8 h/day [9,11], and the World Health Organisation (WHO) recommends a weekly MVPA engagement time of ≥150 min/week [55]. Unless otherwise stated, these classifications were applied both within each intervention group (SBF, LIPA, CON) and globally across all participants to allow for comprehensive comparisons.

### 2.5. Inter-Day Variability Assessment

The coefficient of variation (CV% i.e., (standard deviation/average) × 100) was selected as the primary metric for inter-day reliability, as it is widely used in physical behaviour research to quantify relative variability across days. To account for the limitations in the CV% metric (i.e., mainly not accounting for the number of valid days, sensitivity to outliers, and smaller means being more sensitive to change), we also incorporated individualized variance as a complementary measure. This ensured a more robust assessment of variability, aligning with best practices in the field and providing a more nuanced understanding of day-to-day fluctuations in physical behaviour. Individualised variance was calculated as follows:$$\frac{\sum {\left(x_{i}-\bar{x}\right)}^{2}}{n-1}$$
where S is the sum, *x_i_* is the daily value for that particular physical behaviour parameter, $\bar{x}$ is the weekly mean physical behaviour parameter value for that particular intervention week, and *n* is the number of valid days for that particular intervention week.

### 2.6. Threshold Determination

Inter-day variability thresholds were calculated using the baseline CV% data sample for each PB to examine differences in the likelihood of behavioural compliance. The inter-day PB variability thresholds used a previously published percentile-based classification [20]. Participants were classified as high- or low-regularity, dependent on whether their baseline CV% value for a particular PB was <the 25th percentile sample value (i.e., high regularity) or ≥the 25th percentile (i.e., low regularity) sample value, respectively.

### 2.7. Statistical Analyses

All statistical analyses were carried out using SPSS (Version 26, SPSS Inc., Chicago, IL, USA). Normal distribution and equality of variances between groups were checked using the Shapiro–Wilk and Levene tests, respectively, for all datasets. These were separated into two phases:(1)Changes in the inter-day variability (CV% and individualized variance) in each physical behaviour across time (from baseline to week 8), and the effect of the intervention group (SBF, LIPA, or control). Baseline (raw data and variability) group differences were examined with a one-way analysis of variance (ANOVA) or a Kruskal–Wallis ANOVA (where data were non-normal) given the 3-group design, with post hoc pairwise unpaired student t-test comparisons conducted using the Bonferroni adjustment the or Mann–Whitney U test (for the Kruskal–Wallis ANOVA), respectively. The effects of the interventions were determined using a 2 × 3 split-plot ANOVA [given the study design of 2 time phases (pre- and post-intervention) and 3 intervention groups (SBF, LIPA, and control)] with Bonferroni-corrected post hoc comparisons. In cases of non-normally distributed data, first, within-group comparisons were made using the Wilcoxon signed-rank test, and then between-group pairwise differences were analyzed through a Kruskal–Wallis ANOVA of the relative changes from the baseline [(post-pre)/pre-], with post hoc pairwise comparisons examined by the Mann–Whitney U test. In cases where groups were unmatched at baseline (i.e., significantly different), the baseline values were accounted for in the statistical analysis model as a covariate (ANCOVA analysis).(2)The link between the change in a given physical behaviour metric (e.g., absolute SB time), and its baseline inter-day variability. Here, participants were classified as either high- or low-regularity, and the change (from baseline to week 8) in their absolute physical behaviour outcomes was quantified. The complete set of all baseline classifications can be seen in Table 1. When analysing the PB change by baseline regularity classification, all participants were grouped together regardless of experimental condition (SBF, LIPA, and control). A chi-squared test was used to compare the distribution of inter-day variability classifications (high vs. low regularity) between experimental groups (SBF, LIPA, and CON), sedentary classification groups (ambulatory and couch potato), and activity classification groups (active and inactive) at baseline. These classifications were applied within each group and/or across all participants to ensure balanced comparisons and avoid bias. For a chi-squared test to be appropriate, the expected frequency count for each cell should generally be at least 5. We ensured this assumption was met by verifying that the number of participants in each classification category (e.g., high vs. low regularity) was sufficient for comparison. Specifically, all high-regularity groups had n = 8, meaning comparisons were made against a frequency of n ≥ 8.

The effects of baseline regularity on absolute physical behaviour metric change were determined using a 2 × 2 split-plot ANOVA [2 time phases (pre- and post-intervention) and 2 groups (high baseline regularity vs. low baseline regularity). Similar procedures were also followed for non-parametric data. In cases of non-normally distributed data, within group comparisons were made using the Wilcoxon-Sign Rank test, whilst between group pairwise differences were examined by the Mann-Whitney U test.

Data are reported as Mean ± SD (or Median (IQR) for non-parametric data). Categorical data are reported as proportions (%). Statistical significance was accepted when *p* ≤ 0.05, and statistical trend when *p* was between 0.05 and 0.10 [56]. 95% confidence intervals (CI) were calculated for significant outcomes where data were non-parametric.

## 3. Results

### 3.1. Baseline Differences

All 36 participants completed the study. After excluding participants who did not have a sufficient number of valid days (≥4) for an appropriate monitoring period (n = 5), this left a sample size of n = 31 participants (age, 73 ± 5 y; weight, 66.1 ± 10.1 kg; BMI, 25.8 ± 3.7 kg/m^2^). Importantly, experimental groups were matched at baseline for polypharmacy, FRAT responses, and proportion classified as sedentary/active (see Table 2). The number of participants with 4,5,6, and 7 valid days at baseline was 3, 5, 5, and 18, respectively. This was similar at week 8 (4 valid days, n = 3; 5 valid days, n = 11; 6 valid days, n = 13; 7 valid days, n = 4).

CV% for the 12 PB outcomes ranged from 11–139% (see Table 1). MVPA in bouts of ≥10 min (CV%, 139 ± 145%; individualized variance, 139.3 ± 184.6 min) exhibited the greatest inter-day variability at baseline (shown in Appendix A Appendix A). SB in bouts of <5 min (CV%, 43 ± 20%; individualized variance, 5 ± 5 bouts) exhibited the greatest inter-day variability at baseline of the SB-based outcomes (see Table 1). The proportion of the study sample that therefore qualified as high-regularity (i.e., <25th Percentile) is shown in Table 3, as are the changes in variability at baseline, week 8, with comparisons between groups and time points. Relative SB time exhibited the lowest inter-day variability at baseline (CV%, 11 ± 5%; individualized variance, 38.2 ± 30.8%) (shown in Appendix A Appendix A).

### 3.2. Intervention Outcomes Pertaining to PA Amount

Inter-day variability (CV% and individualized variance) did not significantly change for any PA amount outcome, either across time or dependent on intervention group. Furthermore, no PA amount outcome changed across time when analyzed by baseline regularity classification.

### 3.3. Intervention Outcomes Pertaining to PA Pattern

Importantly, there were no baseline differences between experimental groups (SBF, LIPA, and control), sedentary classification groups (ambulatory and couch potato), or activity classification groups (active and inactive) regarding the number of participants classified as high-/low-regularity for average PA bout length at baseline (*p* > 0.05).

Only average PA bout length significantly changed depending on baseline regularity classification. Specifically, average PA bout length exhibited a significant regularity classification × time effect (*p* = 0.003), and a trend toward a significant time effect (*p* = 0.09) (shown in Figure 1). Participants categorized as high-regularity at baseline for average PA bout length (baseline inter-day variability: <17%) showed a significantly greater increase over time (baseline, 15.8 ± 3.6 min; week 8, 20.8 ± 7.5 min; Δ%, 34 ± 56%) compared to low-regularity participants (baseline, 18.9 ± 4.2 min; week 8, 17.4 ± 4.4 min; Δ%, −7 ± 19%).

Inter-day variability (CV% and individualized variance) did not significantly change for any PA pattern outcome, either across time or by intervention group.

### 3.4. Intervention Outcomes Pertaining to SB Amount

Importantly, there were no baseline differences between experimental groups (SBF, LIPA, and control), sedentary classification groups (ambulatory and couch potato), or activity classification groups (active and inactive), regarding the number of participants classified as high-/low-regularity for proportional SB time at baseline (*p* > 0.05).

When analyzed by baseline regularity classification, proportional SB time exhibited both a significant time effect (*p* = 0.01) and a significant regularity classification × time effect (*p* = 0.049) (shown in Figure 2). Specifically, the participants categorized as high-regularity at baseline for proportional SB time (baseline inter-day variability: < 8%) exhibited a significantly greater reduction over time (baseline, 62.2 ± 6.6%; week 8, 55.0 ± 8.1%; Δ%, −11 ± 11%), when compared to low-regularity participants (baseline, 60.5 ± 7.2%; week 8, 59.5 ± 8.7%; Δ%, −2 ± 10%).

Inter-day variability (CV% and individualized variance) did not significantly change for any SB amount outcome, either across time or dependent on intervention group.

### 3.5. Intervention Results Pertaining to SB Pattern

Inter-day variability (CV% and individualized variance) did not significantly change for any SB amount outcome, either across time or dependent on intervention group. Furthermore, no SB pattern outcome changed across time when analyzed by baseline regularity classification.

## 4. Discussion

This is the first study to investigate inter-day variability and its changes following a sedentary behaviour displacement intervention, in older adults. This study is also the first to investigate any link between baseline inter-day variability and participants’ change in physical behaviour in response to an SB reduction intervention. It was hypothesized that inter-day variability in certain physical behaviours would be marked and would change in response to the SB reduction intervention in this sex and age group.

While our study utilized accelerometer-derived data to quantify physical behaviour, its primary focus was not on methodological validation but rather on understanding how this behaviour fluctuates over time. The validity and reliability of the thigh-mounted triaxial accelerometer algorithms used in this study have been previously established [47,48]. As such, we do not directly compare our findings to other methodological benchmarks, as our aim was to explore the temporal dynamics of sedentary behaviour and physical activity rather than the computational techniques used to derive them.

The first hypothesis was partially upheld since our data showed that inter-day variability (whether expressed as CV% or as individualized variance) differed at baseline PAMVPA%_BL (i.e., MVPA expressed as % of total waking hours PA i.e.,) between the SBF and LIPA groups. Similarly, a number of other variables showed a trend (i.e., STD, STD%, MVPA%, W50%) or a significant change (i.e., PASTD%, ≥10 MVPAbouts) in inter-day variability within group, when comparing baseline to week 8. For the remaining PB variables, however, inter-day variability was unchanged, either as a factor of time or of intervention group. It was also hypothesized that those participants classified as having a high inter-day variability (i.e., low regularity) at baseline would be less likely to alter their behaviour compared to those who were classified as having a low inter-day variability at baseline (i.e., high regularity). Interestingly, participants classified as having a high regularity at baseline for proportional SB time (CV% < 8%) reduced the proportional SB time to a greater extent over time than those classified as having a low regularity at baseline. Similarly, participants classified as having a high regularity at baseline for average PA bout length (CV% < 17%), increased absolute average PA bout length to a greater extent compared to participants categorized as having a low regularity. Therefore, the second hypothesis was upheld, for one PA pattern outcome, and one SB amount outcome.

Moreover, our results suggest that the baseline day-to-day regularity for a given PB affects the likelihood of altering that behaviour in response to an intervention. Most notably, participants classified as having a high regularity at baseline for proportional SB time (CV% < 8%) reduced the proportional amount of SB time to a significantly greater extent (baseline, 62%; week 8, 55%) when compared to low-regularity participants (baseline, 61%; week 8, 60%). Considering a high SB time is associated with poor health in older adults [5,10,11], this is a promising finding given that studies record older adults as spending 63–80% of waking hours in SB (9.4–10.5 h.day) [57] and also given that accumulating <9 h/day sedentary time is associated with a reduced risk of frailty in older adults [6]. Perhaps future intervention studies should focus on achieving a consistent baseline day-to-day regularity (e.g., baseline inter-day variability CV% < 9%) before attempting to reduce absolute SB time, to maximise success with regard to overall behavioural change.

Similarly, participants categorized as having a high regularity at baseline for average PA bout length (baseline inter-day variability: <17%) exhibited a significantly greater increase in bout length over time (~5 min) compared to participants categorized as having a low regularity at baseline (−1 min). Average PA bout length refers to the average length of a PA bout within a given monitoring day for a single participant [48]. Although adults obtain similar health benefits from PA bouts of any length [22,23], longer bouts (>10 min) are predictive of specific health markers such as obesity [58]. Accordingly, older adults expend between 4.1–4.5 kcal/min during light-intensity walking (2.9–3.0 METS) [59]. Considering that the current group of older women on average performed 21 PA bouts per day, increasing each of these by ~5 min could result in an additional 431–473 kcal/day expended (3017–3311 kcal/week).

Contrary to our original hypothesis, inter-day variability did not significantly change, either across time or dependent on intervention group. Previous intervention studies in older adults have found that the intra-day amount and pattern of PA/SB, can be altered in response to an intervention [38,39,40,42]. Our observation of high day-to-day consistency in our group of older women (e.g., 19 ± 6% variability in the total amount of time spent in bouts of physical activity (i.e., LIPA and MVPA)) may point to older adults’ daily routine following a repeated pattern [31] compared to that of younger adults (e.g., 53 ± 19% variability in daily steps (the greatest contributors to which are LIPA and MVPA [43])) [20].

In contrast, MVPA in bouts lasting ≥10 min exhibited substantially low day-to-day regularity at baseline (CV% = 139%) compared to the majority (11/12) of outcomes (11–43%), including MVPA in short bouts of <10 min duration (CV = 21%). This is consistent with previous studies showing that MVPA displays a higher inter-day variability in the elderly compared to younger adults [33, 34], possibly due to the fact that older adults find MVPA substantially challenging and struggle to adhere to such behaviour in the long term [60,61,62]. This supports the recent change to emphasise that ‘150 min can in fact be accumulated in bouts of any length’ [22], including shorter bouts which have greater day-to-day stability according to the current study’s results. It is also promising that the variability in MVPA based outcomes did not change following the current study’s intervention. This demonstrates stability in higher intensities of physical activity when attempting to change physical behaviours like SB and LIPA. Moreover, the average amount of short SB bouts (<5 min) exhibited the second-lowest baseline regularity (CV% 43%) of all 12 outcomes examined, as might be expected considering older adults tend to accumulate SB in a prolonged uninterrupted fashion (i.e., longer bouts) [63], and individuals are largely unaware of how much sedentarism they perform [36].

### 4.1. Study Strengths and Limitations

The major strength of this study is the data collected on inter-day variability (day-to-day regularity) in a group of older women; focusing on this demographic, we aimed to contribute valuable insights specific to a population that is often underrepresented in the scientific literature. Additionally, older women may face unique challenges and considerations related to physical activity and sedentary behaviour, including hormonal changes, caregiving responsibilities, and societal expectations. Another strength is using robust accelerometry techniques to derive detailed 24 h physical behaviour data, including PA amount, PA patterns, SB amount, and SB patterns. The use of 4–7 continuous days of 24 h data collection duration captures temporal variability, provides a reliable sample size for statistical analysis, allows for meaningful longitudinal comparisons, and is feasible within practical constraints. We also quantified individual variance to offset the limitations of the coefficient of variation, mainly not accounting for the number of valid days within a monitoring period, sensitivity to outliers, and smaller means being more sensitive to change.

The major limitation of this study is the use of the <25th percentile threshold to determine whether a participant was classified as having a high or low regularity at baseline. Whilst this was based on the methodology of a previous study [20], the selected threshold is still arbitrary and not based on evidence linking an inter-day variability in the <25th percentile threshold with any specific health outcome. Accordingly, participants whose inter-day variability was only marginally above the <25th percentile (e.g., 26th percentile) would have been classified as having a low regularity based on this binary classification system. To reduce any potential reactivity effects that may have occurred, we used a <4-day exclusion criteria for any monitoring period (at baseline or week 8). Our study is limited by the exclusive recruitment of older women, which affects the generalizability of our findings. Additionally, we acknowledge that our study did not incorporate more detailed psychological measures such as self-efficacy, motivation, or barriers to adherence, which could provide further insight into behavioural consistency and intervention effectiveness. Finally, our observation period was limited to 8 weeks, and longer follow-up periods may be beneficial in assessing sustained changes in sedentary behaviour and physical activity patterns.

### 4.2. Recommendations for Future Work

Future SB reduction studies should use larger sample sizes. Due to a relatively small sample size overall—and especially within the control group—the results of this study should be interpreted cautiously, especially when making generalizations. Considering the high inter-day variability (139%) observed for MVPA (in bouts of ≥10 min), interventions seeking to increase MVPA in older adults should take the potential low consistency of this behaviour into account. It may also be interesting to observe how intervening through manipulating baseline inter-day variability affects behavioural change (i.e., stabilizing baseline SB to more successfully reduce SB). Future research should also explore psychological determinants of adherence, such as motivation, self-efficacy, and perceived barriers, to better understand their role in sustaining long-term behavioural change. Moreover, extending study durations beyond 8 weeks would allow for a more comprehensive evaluation of long-term intervention effects and variability in physical activity patterns over time.

More generally, future cross-sectional investigations should place greater emphasis on inter-day variability as a credible PB metric. This could be done through investigating more varied thresholds (e.g., <10th vs. ≥90th percentile), different populations (e.g., younger adults), and how they are associated with different aspects of health (e.g., cardiometabolic health).

## 5. Conclusions

Day-to-day regularity of PB has hitherto been overlooked as a valuable metric in both physical activity research and policy messaging. Our results show that whilst inter-day variability did not change over time, baseline regularity classification (high vs. low) was linked to certain changes in physical behaviour, in response to an intervention of reduced SB/increased LIPA in older women. Ergo, participants who performed SB (% time) and/or PA (bout length) with high regularity pre-intervention exhibited greater reduced SB time/increased PA bout length post-intervention. Establishing day-to-day consistency in the pattern of one’s PB may help to achieve beneficial behavioural change in response to a lifestyle intervention.

These findings suggest that greater day-to-day baseline regularity in PB may enhance responsiveness to behavioural interventions in older adults. This study supports the notion that individuals with regular physical behaviour may better adhere to and benefit from structured behavioural interventions, aligning with broader theories of habit formation and self-regulation. Additionally, these results could inform the design of personalised physical activity interventions, potentially benefitting others populations beyond older adults, including those with chronic conditions or younger adults, by considering baseline behavioural regularity as a predictor of intervention success.

## Figures and Tables

**Figure 1 sensors-25-02194-f001:**
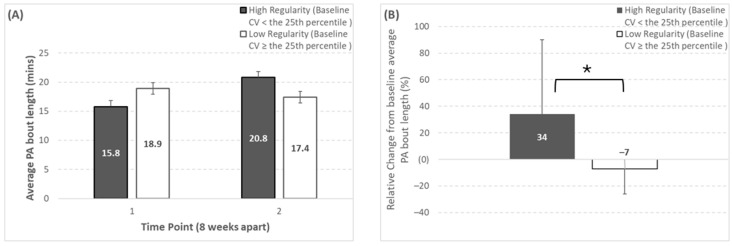
Changes over time for average physical activity bout length depending on baseline regularity classification (regularity classification thresholds can be seen in Table 1). Panels (**A**,**B**) represent the absolute and relative change, respectively. * Represents a significant group × time effect (*p* = 0.003) from the Mann–Whitney U test comparing the relative change from baseline between the two regularity groups. Note that the number of participants (n) in the high-regularity group at baseline was 7, 7, 1 for SBF, LIPA, and control, respectively (corresponding to 58%, 54%, and 17% of those groups, or 48% of the whole study sample).

**Figure 2 sensors-25-02194-f002:**
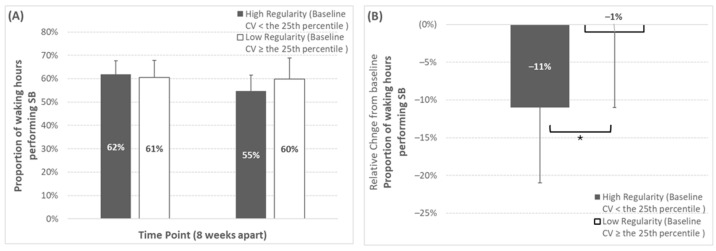
Changes over time for proportional sedentary behaviour time depending on baseline regularity classification (regularity classification thresholds can be seen in Table 1). Panels (**A**,**B**) represent the absolute and relative change, respectively. * Represents a significant group × time effect (*p* = 0.049) from the two-way mixed design ANOVA. Note that the number of participants (n) in the high-regularity group at baseline was 5, 5, 2 for SBF, LIPA, and control, respectively (corresponding to 42%, 38%, and 33% of those groups, or 39% of the whole study sample), for this outcome variable.

**Table 1 sensors-25-02194-t001:** Inter-day variability classifications at baseline for physical behaviour (PB) variables (n = 31) *.

Physical Behaviour Category	Average Baseline Absolute PB Outcome	Average Baseline Inter-Day Variability Expressed as Coefficient of Variation (%)	Low Regularity Threshold Value (≥25th Percentile Value)	Average Baseline Inter-Day Variability Expressed as Individualized Variance
**PA amounts**				
Standing Time (hours)	1.3 ± 0.5	28 ± 8%	≥23%	0.1 ± 0.1
Light-intensity physical activity time (hours)	2.1 ± 0.4	21 ± 10%	≥17%	0.2 ± 0.2
Moderate-to-vigorous physical activity time (hours)	2.7 ± 0.8	20 ± 6%	≥17%	0.2 ± 0.3
MVPA in bouts ≥ 10 min duration (minutes)	8.5 ± 11.9	139 ± 145%	≥55%	139.3 ± 184.6
MVPA in bouts of <10 min duration (minutes)	150 ± 44	21 ± 8%	≥18%	658 ± 870
Daily sum of PA bout time (minutes)	361 ± 73	19 ± 7%	≥15%	3400 ± 3333
**PA patterns**				
Bouts of PA (number)	21 ± 5	18 ± 10%	≥15%	13 ± 10
Average Daily PA bout length (minutes)	18.1 ± 4.2	25 ± 17%	≥17%	16 ± 38
**SB amounts**				
Sedentary Behaviour Time (hours)	9.5 ± 1.1	12 ± 8%	≥9%	1.2 ± 1.4
Sedentary Behaviour Time (% of waking hours)	60.8 ± 7.0	11 ± 5%	≥8%	38.2 ± 30.8
**SB patterns**				
SB in bouts of <5 min (number)	6 ± 2	43 ± 20%	≥14%	5 ± 5
Average Daily Sedentary Behaviour Bout Length (Minutes)	32.9 ± 9.2	27 ± 17%	≥20%	70.7 ± 158.2

* Data are presented as mean ± SD (non-parametric data are presented as median ± interquartile range). The third column ‘Low Regularity Threshold Value (≥25th percentile value)’ is the CV% value for each variable representing the 25th percentile within the sample, and the threshold for determining ‘low regularity’. LIPA, Light-intensity physical activity; MVPA, moderate-to-vigorous physical activity; PA, physical activity; SB, sedentary behaviour; SD, standard deviation. Baseline is Week 0.

**Table 2 sensors-25-02194-t002:** Participant characteristics, physical activity, and SB outcomes at baseline (n = 31) *. Bold highlights for non-normally distributed data (95% confidence in brackets indicates lower and upper bounds). There were no significant main effects of group type.

	SBF (n = 12)	LIPA (n = 13)	Control (n = 6)	ANOVA (or * Kruskal–Wallis)
**Age (y)**	74 ± 5	73 ± 5	**70 ± 3 (67, 73)**	***p* = *n.s.***
Weight (kg)	68.6 ± 11.3	65.5 ± 8.6	65.4 ± 9.7	*p* = *n.s.*
BMI (kg·m^2^)	26.9 ± 3.6	25.3 ± 3.6	26.2 ± 3.7	*p* = *n.s.*
**FRAT responses (n)**	**1.0 ± 0.9 (0.4, 1.6)**	**0.7 ± 0.8 (0.2, 1.1)**	**0.5 ± 0.8 (−0.4, 1.4)**	***p* = *n.s.***
**Polypharmacy (n)**	**2.0 ± 2.0 (0.8, 3.2)**	**0.7 ± 1.1 (0.2, 1.4)**	1.7 ± 2.0	***p* = *n.s.***
**PA amounts**				
Standing time (hours)	1.16 ± 0.57	1.42 ± 0.57	1.25 ± 0.39	
Light-intensity physical activity time (hours)	2.19 ± 0.48	2.08 ± 0.35	2.23 ± 0.7	*p* = *n.s.*
Moderate-to-vigorous physical activity time (hours)	2.95 ± 0.96	2.5 ± 0.78	3.6 ± 1.09	*p* = *n.s.*
**MVPA in bouts ≥ 10 min duration (minutes)**	**21.4 ± 25.5 *(6.7, 36.1)***	**7.8 ± 5.5 *(4.7, 11.0)***	**15.9 ± 13.8 *(1.4, 30.5)***	***p* = *n.s.***
MVPA in bouts of <10 min duration (minutes)	145 ± 34	140 ± 47	181 ± 49	*p* = *n.s.*
Daily sum of PA bout time (minutes)	369.77 ± 68.26	351.9 ± 77.04	416.03 ± 120.64	*p* = *n.s.*
Standing (%waking hours)	7.28 ± 3.66	9.04 ± 3.51	8.13 ± 2.5	*p* = *n.s.*
LIPA (%waking hours)	13.76 ± 2.93	13.32 ± 2.06	14.45 ± 4.04	*p* = *n.s.*
MVPA (%waking hours)	18.53 ± 5.96	15.93 ± 4.53	23.44 ± 6.92	*p* = *n.s.*
**PA Patterns**				
**Bouts of PA (number)**	**21.47 ± 4.9 *(18.6, 24.3)***	**21.24 ± 3.3 *(19.3, 23.1)***	**21.41 ± 3.68 *(16.5, 25.0)***	***p* = *n.s.***
Average daily PA bout length (minutes)	18.65 ± 4.73	17.12 ± 3.59	20.77 ± 6.67	*p* = *n.s.*
SBPAtime (mins)	1.46 ± 0.55	1.23 ± 0.53	1.57 ± 0.46	*p* = *n.s.*
STDPAtime (mins)	18.84 ± 9.08	23.55 ± 6.85	17.62 ± 3.8	*p* = *n.s.*
LIPAPAtime (mins)	35.01 ± 5.99	35.27 ± 5.12	31.13 ± 3.99	*p* = *n.s.*
MVPAPAtime (mins)	44.68 ± 10.11	39.95 ± 7.03	49.68 ± 6.19	*p* = *n.s.*
>10 MINSMVPABOUTS (n is count)	1.06 ± 1.19	0.54 ± 0.36	0.77 ± 0.66	*p* = *n.s.*
SPMVPA (mins)	155.78 ± 42.52	142.41 ± 46.28	202.56 ± 71.51	*p* = *n.s.*
**SB Amounts**				
Sedentary behaviour time (hours)	9.64 ± 1.24	9.63 ± 1.05	8.3 ± 1.77	*p* = *n.s.*
Sedentary behaviour time (% of waking hours)	60.43 ± 7.01	61.71 ± 7.3	53.98 ± 12.06	*p* = *n.s.*
Sleep (hrs)	8.06 ± 0.87	8.37 ± 0.65	8.62 ± 0.78	*p* = *n.s.*
>5 MinSB (mins)	16.17 ± 3.13	15.87 ± 1.55	15.62 ± 2.86	*p* = *n.s.*
MeanSB (mins)	32.6 ± 10.16	32.97 ± 8.23	27.61 ± 10.29	*p* = *n.s.*
**SB Patterns**				
**SB in bouts of <5 min (number)**	**6.12 ± 2.2 *(4.9, 7.4)***	**6.2 ± 2.43 *(4.8, 7.6)***	**6.56 ± 2.56 *(3.2, 9.4)***	***p* = *n.s.***
Average daily sedentary behaviour Bout length (minutes)	** *32.2 (27.7–34.6)* **	** *31.9 (27.4–39.4)* **	** *28.2 (21.4–36.8)* **	*p* = *n.s.*
SBBreak (n is count)	21.47 ± 4.9	21.25 ± 3.31	21.41 ± 3.68	*p* = *n.s.*
Alpha-scaling parameter sedentary bout length distribution	1.44 ± 0.04	1.45 ± 0.04	1.47 ± 0.06	*p* = *n.s.*
W50 (%)-half of total SB is accumulated in SB bouts ≤ this duration	59.64 ± 15.35	58.6 ± 13.07	45.96 ± 13.68	*p* = *n.s.*
**Participants Categorisation**				
Proportion meeting 150 min/week MVPA (%)	16%	0%	0%	*p* = *n.s. (chi-square test)*
Proportion classified as sedentary (%)	100%	100%	84%	*p* = *n.s. (chi-square test)*

* Data are presented as mean ± SD [non-parametric data are presented as median (interquartile range)]. After excluding participants who did not have a sufficient number of valid days (≥4) for an appropriate monitoring period (n = 5), this left a sample size of n = 31 participants (SBF, n = 12; LIPA, n = 13; control, n = 6). BMI, body mass index; FRAT, falls risk assessment tool; LIPA, light-intensity physical activity; MVPA, moderate-to-vigorous physical activity; PA, physical activity; SB, sedentary behaviour; SD, standard deviation; n.s., non-significant.

**Table 3 sensors-25-02194-t003:** Proportion of participants by intervention group, with high regularity in key physical behaviour outcomes. Data shown are %. Bold highlights for non-normally distributed data (95% confidence in brackets indicate lower and upper bounds). All data showed homogeneous variance. (A) At baseline. A single significant difference was found and this was in PAMVPA%_BL (*p* = 0.041), which indicated at post hoc the difference being solely between LIPA and SBF groups (mean difference = 4.9%, *p* = 0.039). (B) At Week 8, all the variability data were non-normally distributed. Kruskal–Wallis tests found no intervention group difference in the group variability in PB outcomes. (C) Wilcoxon signed-rank tests for the change in individual participants’ variability within each group. Results are one-tailed exact *p* values. Statistical trends (*p* < 0.1) and significance (*p* ≤ 0.05) are indicated. n.s., non-significant.

**(A) Proportion of High Regularity at Baseline**	**SBF**	**LIPA**	**Control**	**ANOVA (* Kruskal–Wallis)**	**Whole Sample**
**PA amounts**					
**Standing time (hours)**	42%	**46% (24%, 37%)**	17%	* ***p* = *n.s.***	39%
**Light-intensity physical activity -LIPA time (hours)**	75%	46%	67%	*p* = *n.s.*	61%
**Moderate-to-vigorous physical activity -MVPA time (hours)**	83%	**77% (19%, 28%)**	100%	* ***p* = *n.s.***	84%
**Standing (%waking hours)**	42%	38%	17%	*p* = *n.s.*	35%
**LIPA (%waking hours)**	75%	46%	67%	*p* = *n.s.*	61%
MVPA (%waking hours)	83%	69%	100%	*p* = *n.s.*	81%
MVPA in bouts ≥ 10 min duration (minutes)	33%	15%	0%	*p* = *n.s.*	19%
**PA patterns**					
Bouts of physical activity -PA (number)	83%	85%	67%	*p* = *n.s.*	81%
Physical activity bouts (Mins)	67%	77%	100%	*p* = *n.s.*	77%
Average daily PA bout length (minutes)	58%	54%	33%	*p* = *n.s.*	52%
PASTD%_BL	83%	69%	83%	*p* = *n.s.*	77%
PAMVPA%_BL	100%	100%	100%	*p* = 0.041	100%
**SB amounts**					
**Sedentary behaviour time (hours)**	**100% (9%, 16%)**	100%	100%	* ***p* = *n.s.***	100%
Sedentary behaviour time (% of waking hours)	100%	100%	100%	*p* = *n.s.*	100%
SB shorter than 5 min (mins)	8%	15%	17%	*p* = *n.s.*	13%
**SB longer than 5 min (mins)**	75%	85%	**100% (15%, 25%)**	* ***p* = *n.s.***	84%
**Mean SB bout duration (Mins)**	42%	31%	**83% (17%, 31%)**	* ***p* = *n.s.***	45%
**SB patterns**					
Breaks in sedentary behaviour	83%	85%	67%	*p* = *n.s.*	81%
**Alfa-scaling parameter sedentary bout length distribution**	100%	100%	**100% (3%, 5%)**	* ***p* = *n.s.***	100%
W50%-half of total SB is accumulated in SB bouts ≤ this duration	42%	15%	17%	*p* = *n.s.*	26%
**(B) Proportion of high regularity at Week 8**	**SBF**	**LIPA**	**Control**	**Kruskal–Wallis**	**Whole Sample**
**PA amounts**					
**Standing time (hours)**	**50%**	**69%**	**67%**	***p* = *n.s.***	**61%**
**Light-intensity physical activity -LIPA time (hours)**	**75%**	**54%**	**50%**	***p* = *n.s.***	**61%**
**Moderate-to-vigorous physical activity-MVPA time (hours)**	**67%**	**46%**	**67%**	***p* = *n.s.***	**58%**
**Standing (%waking hours)**	**50%**	**69%**	**50%**	***p* = *n.s.***	**58%**
**LIPA (%waking hours)**	**75%**	**62%**	**50%**	***p* = *n.s.***	**65%**
**MVPA (%waking hours)**	**67%**	**46%**	**67%**	***p* = *n.s.***	**58%**
**MVPA in bouts ≥10 min duration (minutes)**	**58%**	**8%**	**17%**	***p* = *n.s.***	**29%**
**PA patterns**					
**Bouts of physical activity-PA (number)**	**83%**	**62%**	**83%**	***p* = *n.s.***	**74%**
**Physical activity bouts (Mins)**	**83%**	**69%**	**67%**	***p* = *n.s.***	**74%**
**Average daily PA bout length (minutes)**	**50%**	**62%**	**50%**	***p* = *n.s.***	**55%**
**PASTD%_BL**	**83%**	**77%**	**83%**	***p* = *n.s.***	**81%**
**PAMVPA%_BL**	**92%**	**100%**	**100%**	***p* = *n.s.***	**97%**
**SB amounts**					
**Sedentary behaviour time (hours)**	**83%**	**92%**	**100%**	***p* = *n.s.***	**90%**
**Sedentary behaviour time (% of waking hours)**	**83%**	**92%**	**100%**	***p* = *n.s.***	**90%**
**SB shorter than 5 min (mins)**	**8%**	**0%**	**17%**	***p* = *n.s.***	**6%**
**SB longer than 5 min (mins)**	**50%**	**92%**	**67%**	***p* = *n.s.***	**71%**
**Mean SB bout duration (Mins)**	**42%**	**8%**	**50%**	***p* = *n.s.***	**29%**
**SB patterns**					
**Breaks in sedentary behaviour**	**83%**	**62%**	**83%**	***p* = *n.s.***	**74%**
**Alfa-scaling parameter sedentary bout length distribution**	**100%**	**100%**	**100%**	***p* = *n.s.***	**100%**
**W50%-half of total SB is accumulated in SB bouts ≤ this duration**	**25%**	**31%**	**33%**	***p* = *n.s.***	**29%**
**(C) Change in participants’ variability at baseline vs. week 8**	**SBF**	**LIPA**	**Control**	**Whole Sample**	
**PA amounts**					
**Standing time (hours)**	***p* = *n.s.***	***p* = 0.084**	***p* = *n.s.***	***p* = *0.091***	
**Light-intensity physical activity-LIPA time (hours)**	***p* = *n.s.***	***p* = *n.s.***	***p* = *n.s.***	***p* = *n.s.***	
**Moderate-to-vigorous physical activity -MVPA time (hours)**	***p* = *n.s.***	***p* = *n.s.***	***p* = *n.s.***	***p* = *0.079***	
**Standing (%waking hours)**	***p* = *n.s.***	***p* = *0.064***	***p* = *n.s.***	***p* = *0.079***	
**LIPA (%waking hours)**	***p* = *n.s.***	***p* = *n.s.***	***p* = *n.s.***	***p* = *n.s.***	
**MVPA (%waking hours)**	***p* = *0.055***	***p* = *0.073***	***p* = *n.s.***	***p* = 0.029**	
**MVPA in bouts ≥10 min duration (minutes)**	***p* = *n.s.***	***p* = *n.s.***	***p* = *0.031***	***p* = *n.s.***	
**PA patterns**					
**Bouts of physical activity-PA (number)**	***p* = *n.s.***	***p* = *n.s.***	***p* = *n.s.***	***p* = *n.s.***	
**Physical activity bouts (Mins)**	***p* = *n.s.***	***p* = *n.s.***	***p* = *n.s.***	***p* = *n.s.***	
**Average daily PA bout length (minutes)**	***p* = *n.s.***	***p* = *n.s.***	***p* = *n.s.***	***p* = *n.s.***	
**PASTD%_BL**	***p* = *n.s.***	***p* = 0.020**	***p* = *n.s.***	***p* = *n.s.***	
**PAMVPA%_BL**	***p* = *n.s.***	***p* = *n.s.***	***p* = *n.s.***	***p* = *n.s.***	
**SB amounts**					
**Sedentary behaviour time (hours)**	***p* = *n.s.***	***p* = *n.s.***	***p* = *n.s.***	***p* = *n.s.***	
**Sedentary behaviour time (% of waking hours)**	***p* = *n.s.***	***p* = *n.s.***	***p* = *n.s.***	***p* = *n.s.***	
**SB shorter than 5 min (mins)**	***p* = *n.s.***	***p* = *n.s.***	***p* = *n.s.***	***p* = *n.s.***	
**SB longer than 5 min (mins)**	***p* = *n.s.***	***p* = *n.s.***	***p* = *n.s.***	***p* = *n.s.***	
**Mean SB bout duration (Mins)**	***p* = *n.s.***	***p* = *n.s.***	***p* = *n.s.***	***p* = *n.s.***	
**SB Patterns**					
**Breaks in sedentary behaviour**	***p* = *n.s.***	***p* = *n.s.***	***p* = *n.s.***	***p* = *n.s.***	
**Alfa-scaling parameter sedentary bout length distribution**	***p* = *n.s.***	***p* = *n.s.***	***p* = *n.s.***	***p* = *n.s.***	
**W50%_half of total SB is accumulated in SB bouts ≤ this duration**	***p* = *0.055***	***p* = *n.s.***	***p* = *n.s.***	***p* = *n.s.***	

## Data Availability

The raw data supporting the conclusions of this article will be made available on request, by the first or corresponding authors, without undue reservation.

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
