# Peer review of "Daily Variability in Sedentary Behaviour and Physical Activity Responsiveness in Older Women"

_sensors, 2025, doi:10.3390/s25072194_

Round 1
Reviewer 1 Report (Previous Reviewer 1)
Comments and Suggestions for Authors
Thank you for your work on both the original submission as well as the revised manuscript. The clarifications and responses to the editorial commentary, as well as the in-text changes/additions to the manuscript sufficiently addressed all questions posed.
Author Response
We thank our reviewer 1 for their support and in helping to improve our manuscript.
Reviewer 2 Report (New Reviewer)
Comments and Suggestions for Authors
I would like to thank you for the opportunity to review the submitted work entitled “The day-to-day variability in physical behaviour in older 2 women: comparative impact upon sedentary behaviour fragmentation vs. single-bout physical activity responsiveness”.
The topic of the article is current and important because a sedentary lifestyle poses a significant health risk to older people, and the effectiveness of various interventions in this population group is still poorly understood.
The work is very well prepared in every respect and deserves praise.
Minor comments:
Despite the sample size being based on other publications, the number of participants was relatively low.
Study limitations include the lack of analysis of psychological measures of adherence. The authors did not assess participants' self-efficacy, motivation, or barriers to adherence.
Additionally, observation was not continued beyond week 8.
Author Response
1. Summary |
|
|
Thank you for your insightful comments and for recognising the strengths of our work. We appreciate your suggestions regarding sample size, psychological measures of adherence, and the observation period. While addressing these aspects was beyond the scope of our current study, we have incorporated them into the ‘Study Limitations’ and ‘Future Work’ sections to acknowledge their importance and guide further research in this area. We are grateful for your thorough review and constructive feedback. Please find the detailed responses below and the corresponding revisions/corrections highlighted/in track changes in the re-submitted files.
|
||
Comments 1: Minor comments: Despite the sample size being based on other publications, the number of participants was relatively low. Study limitations include the lack of analysis of psychological measures of adherence. The authors did not assess participants' self-efficacy, motivation, or barriers to adherence. Additionally, observation was not continued beyond week 8.
|
||
Response 1: Study Strengths and Limitations (Revised i.e. added text): Additionally, we acknowledge that our study did not incorporate more detailed psychological measures such as self-efficacy, motivation, or barriers to adherence, which could provide further insight into behavioural consistency and intervention effectiveness. Finally, our observation period was limited to 8 weeks, and longer follow-up periods may be beneficial in assessing sustained changes in sedentary behaviour and physical activity patterns.
Recommendations for Future Work (Revised i.e. added text): Future research should also explore psychological determinants of adherence, such as motivation, self-efficacy, and perceived barriers, to better understand their role in sustaining long-term behaviour change. Moreover, extending study durations beyond 8 weeks would allow for a more comprehensive evaluation of long-term intervention effects and variability in physical activity patterns over time. |
Reviewer 3 Report (New Reviewer)
Comments and Suggestions for Authors
See attached document

See the attached document. Small errors that are normal but should be corrected
Author Response
Response to Reviewer 3 Comments
|
||
1. Summary |
|
|
Thank you very much for taking the time to review this manuscript. Please find the detailed responses below and the corresponding revisions/corrections highlighted/in track changes in the re-submitted files.
|
||
2. Point-by-point response to Comments and Suggestions for Authors |
|
|
Comments 1: The English could be improved to more clearly express the research.
|
||
Response 1: Thank you for pointing this out. We have combed through the text and corrected any unclear statements.
|
||
Comments 2: (a) Can be improved: (i) Does the introduction provide sufficient background and include all relevant references?; (ii) Is the research design appropriate? (iii) Are the conclusions supported by the results? (b) Must be improved (i) Are the methods adequately described?; (ii) Are the results clearly presented? |
||
Response 2: Thank you for your assessments. We have addressed the points by points made in the separate document (please see below) |
||
|
||
3. Points made in a separate word document- |
General comments sensors-3471269:
RESPONSE: We thank you for your helpful suggestions. Point by point responses are below.
Abstract:
➢ Comment 3: Consider clarifying the central hypothesis further. While the relationship between inter-day variability and intervention responsiveness is mentioned, more precise articulation of the study's objectives might improve clarity.
Response 3- We have revised the abstract to explicitly state our hypotheses, improving clarity on the study's objectives.
Introduction:
➢ Comment 4: The introduction is strong but would benefit from elaborating on why the specific focus on older adults is critical, as this may clarify the study's relevance in the broader context of sedentary behavior interventions.
Response 4- We have expanded on the rationale for focusing on older individuals.
Methodology:
➢ Comment 5: It is important to specify the reasons for the unequal sample sizes across groups (SBF: n=14, LIPA: n=14, Control: n=8). Was the control group smaller due to data exclusions or another reason? A clarification is necessary to avoid potential bias concerns.
Response 5-The control group was smaller than initially planned due to drop out upon allocation to this group. We have also added a reference to the literature with regards to group attrition.
Statistical Analysis:
➢ Comment 6: While statistical tests are appropriately chosen, the paper would benefit from a more precise explanation of why these particular tests were chosen, especially when normality assumptions might be questionable in specific datasets. It is good that non-parametric tests are used where appropriate, but ensure that all statistical methods are explained sufficiently for readers unfamiliar with advanced analysis techniques.
Response 6- explanatory text has now been added to the description of this section to ensure that it is clear to a larger audience.
➢ Comment 7: Chi-Squared Tests: The chi-squared test used for proportions of participants classified as sedentary needs a justification. The assumption of expected frequency counts should be checked.
Response 7- The Chi-squared test was used to compare the distribution of baseline inter-day variability classifications (high vs. low regularity) across the experimental groups (SBF, LIPA, and control), as well as the sedentary and activity classification groups. Given our overall sample size), we checked that the expected frequency counts in each of the comparison categories were sufficient for the test to be valid.
For a Chi-squared test to be appropriate, the expected frequency count for each cell should generally be at least 5. We ensured this assumption was met by verifying that the number of participants in each classification category (e.g., high vs. low regularity) was sufficient for comparison. Specifically, all high-regularity groups had n=8, meaning comparisons were made against a frequency of n≥8. If any category had an expected frequency below 5, we would have used alternative methods, such as Fisher’s Exact Test, to account for small sample sizes. This clarification has now been added to the text.
➢Comment 8: Sample Size: The small control group (n=8) raises questions regarding statistical power. A power analysis could be included to justify this sample size or suggest limitations regarding the generalization of the results. Include effect sizes (see below).
Response 8 – Regarding our reviewer’s concern regarding the small control group (n=8). A power analysis was not conducted due to the novelty of investigating changes in the inter-day variability of physical behaviour in older women. To our knowledge no similar previous investigation exists that can be drawn upon for a power analysis for a ‘sufficient’ sample size for each group, including control. Ultimately, we believe this is a key strength of the current manuscript through providing not only completely novel baseline physical behaviour regularity data, but also the change over an intervention period. Nevertheless, this has now been acknowledged within the original text
‘An a priori power analysis was not performed due to lack of previous investigations examining changes in physical behaviour regularity in older adults, highlighting the novelty of the current study. As such, estimation of required sample size to detect significant changes in the outcomes relevant to older adults physical function (i.e. gait speed) was based on previous SB interventions in older adults that utilized ~25–38 participants (Barone Gibbs et al., 2017; Harvey et al., 2018; Rosenberg et al., 2015). The current achieved sample size of 31-36 older women, falls well within this range.’
We have also addressed this more explicitly within the potential limitations section towards the end of the manuscript:
‘Future SB reduction studies should use greater sample sizes. Due to a relatively small sample size overall, but especially within the control group the results of this study should be interpreted cautiously especially when making generalisations.’
Finally, on effect sizes: We acknowledge our reviewer’s request for effect sizes. However, given the small sample size, the calculation of effect sizes may not provide meaningful or reliable estimates of practical significance. As we have already noted, the study’s generalizability is limited, and we have been transparent about this throughout the manuscript. Additionally, including effect sizes for all comparisons would overcomplicate the results tables without necessarily adding valuable insight, particularly for non-significant findings. Our current approach prioritizes clarity while ensuring that key results remain the focus. We believe this is the most appropriate way to present the data given the study’s scope and sample constraints.
Results:
➢ Comment 9: Some results show p-values greater than 0.05, which are interpreted as non-significant (n.s.), please include the actual p-value.
Response 9 - We appreciate your observation regarding the p-values. In the manuscript, there are 141 instances of "n.s.". We appreciate the reviewer’s request for exact p-values and effect sizes. However, given the small sample size, we have already acknowledged the limitations regarding generalizability throughout the manuscript. Reporting exact p-values for all comparisons, including non-significant ones, would not provide meaningful additional insights, given the limited statistical power.
To maintain clarity and readability, we have opted to report exact p-values only when p < 0.1, while denoting non-significant values as p n.s.. This approach avoids overloading the results tables with excessive numerical detail, ensuring that key findings remain the focus. Given the inherent constraints of the sample size, we believe this is the most appropriate and transparent way to present the data.
➢ Comment 10: Adding more context, such as effect sizes, would be helpful to allow readers to gauge the strength of the observed trends.
Response 10 – Please see our previous response (response 8).
➢ Comment 11: The study's focus on variability is well-documented, but consider clarifying whether this measure aligns with existing standards in the field.
Response 11 - We acknowledge the need for clarity regarding the measure of variability. To the authors knowledge there is no ‘existing standard’ that exists in the field for measuring variability of physical behaviours like sedentary behaviour or physical activity. We did acknowledge this within the original manuscript’s introduction. However, we have now made this more explicit within the introduction (new text in yellow):
‘Inter-day variability is a recognized method of monitoring athletes’ training load [25-27]. However, there is no existing standard for measuring variability of physical behaviour in healthy populations. Previous studies have examined the variability of physical behaviour in older clinical populations, by examining the influence of week vs. weekend day [XX]. Interestingly, a previous study, found that adults with the greatest regularity of daily steps (inter-day co-efficient of variation ≤40%) reduced their weight by 0.33kg more, compared to those with the lowest daily steps regularity (CV ≥ 63%) [20], which suggests that PA variability may be a biomarker for the impact of PA on health [20].
New references added above ([XX]) are:
- Schwendinger, F., Wagner, J., Infanger, D., Schmidt-Trucksäss, A. and Knaier, R., 2021. Methodological aspects for accelerometer-based assessment of physical activity in heart failure and health. BMC Medical Research Methodology, 21, pp.1-12.
- Abel, B., Pomiersky, R., Werner, C., Lacroix, A., Schaeufele, M. and Hauer, K., 2019. Day-to-day variability of multiple sensor-based physical activity parameters in older persons with dementia. Archives of gerontology and geriatrics, 85, p.103911.
In the revised manuscript, we now explicitly state how this measure aligns with existing standards in the field and justify its use in the context of this study (see section 2.5- Inter-Day Variability Assessment).
➢ Comment 12: Some of the CV% values reported are high (39% for MVPA), and it might help to explain why such variability is expected or what it indicates regarding intervention outcomes.
Response 12 – The authors cannot locate a CV% figure related to an MVPA variable of 39%. The reviewer may be referring to the high CV% value of 139% for MVPA, in bouts ≥10 minutes at baseline, which is indeed substantially larger than the majority of CV% outcomes presented. Whilst we did acknowledge this in the manuscript, the reviewer is correct that such high values were not sufficiently explained regarding expectations of such large data or intervention outcomes. As such, the following text has been added into the discussion section:
‘In contrast, MVPA in bouts lasting ≥10 minutes, exhibited substantially low day-to-day regularity at baseline (CV% = 139%), compared to the majority (11/12) of outcomes (11-43%), including MVPA in short bouts of <10 minutes duration (CV = 21%). This is consistent with previous studies showing that MVPA displays a higher inter-day variability in the elderly compared to younger adults [XX, XX], possibly due to the fact that older adults find MVPA substantially challenging and struggle to adhere to such behaviour in the long term [XX, XX]. This supports the recent change to emphasise that ‘150 minutes can in fact be accumulated in bouts of any length’ [XX], including shorter bouts which appear to have greater day-to-day stability according to the current study’s results. It is also promising that the variability in MVPA based outcomes did not change following the current study’s intervention. This demonstrates stability in higher intensities of physical activity, when attempting to change physical behaviours like SB and LIPA.’
New references added above:
- Wickel, E.E. and J.C. Eisenmann, Within-and between-individual variability in estimated energy expenditure and habitual physical activity among young adults. European journal of clinical nutrition, 2006. 60(4): p. 538-544.
- Ridley, K., et al., Intra-individual variation in children's physical activity patterns: Implications for measurement. Journal of science and medicine in sport, 2009. 12(5): p. 568-572.
- Brawner CA, Churilla JR, Keteyian SJ. Prevalence of physical activity is lower among individuals with chronic disease. Medicine and science in sports and exercise. 2016;48(6):1062-7.
- Jiménez-Pavón D, Carbonell-Baeza A, Lavie CJ. Physical exercise as therapy to fight against the mental and physical consequences of COVID-19 quarantine: Special focus in older people. Progress in Cardiovascular Diseases. 2020
- Nobles, J., et al., “Let’s talk about physical activity”: understanding the preferences of under-served communities when messaging physical activity guidelines to the public. International journal of environmental research and public health, 2020. 17(8): p. 2782.
➢ Comment 13: The results are well-supported by figures and tables, but some figures could benefit from more straightforward captions explaining the significance of the data for each group. A better explanation of the statistical measures in the figure legends would enhance readability. (see below)
Response 13 - Thank you for the feedback on the figures and tables. We have revised the captions to include clearer explanations of the significance of the data for each group and provide more detailed descriptions of the statistical measures in the figure legends to enhance readability. Please see responses below.
Tables:
➢ Comment 14: P Values (instead of "n.s."): Simply using "n.s." (not significant) can be unhelpful, especially in a scientific context where p-values provide more detailed information. I recommend replacing "n.s." with the exact p-value, such as:"p = 0.12" instead of "n.s." This provides more transparency, and readers can understand the degree of significance (or lack thereof), which is crucial for interpreting the results accurately.
Response 14 – Please refer to our earlier response (response 9).
➢ Comment 15: Effect Sizes: Adding effect sizes to the tables will make the results more interpretable and allow readers to understand not only whether a difference is statistically significant but also the magnitude of the effect. Effect size measures can be added in a separate column or in parentheses alongside the p-values, depending on the format that fits best with your table design.
Response 15 – Please refer to our earlier response (response 8).
➢ Comment 16: Simplification of Table Layout: Some tables may appear overwhelming due to the number of categories or outcomes listed. Breaking these tables into smaller, more focused tables could enhance readability. This avoids clutter and makes the findings easier to digest.
Response 16 - We appreciate the suggestion to simplify the table layout. However, the request to produce more comprehensive tables was a request from a previous reviewer to avoid readers having to interpret data across multiple tables. We have therefore kept the detailed tables.
➢ Comment 17: Clarification of Units and Statistical Terminology: Ensure that units of measurement are consistently stated in the table headers, footnotes, or labels. For example, if the values represent hours, minutes, or percentages, this should be clarified in the header or a note at the bottom of the table. For statistical terms, clarify terms such as "SD" (standard deviation) and "IQR" (interquartile range).
Response 17 - Thank you for pointing out the need for clarity in units and statistical terminology. Within table 3, we have now clearly labelled all outcome variables in as much detail as possible. After checking over all tables, we are confident there is in-fact consistent labelling of units (e.g., hours, minutes, percentages). To the reviewers second point the statistical term ‘SD’ was not defined as an abbreviation in the table footnote, so this has been added to all relevant tables.
Furthermore, we have made additional attempt to clarify any potential confusion within tables by providing additional explanations where necessary. For example the following text has been added into the footnote of table 1:
‘The third column named ‘Low Regularity Threshold Value (≥ 25th percentile value)’ is the CV% value for each variable representing the 25th percentile within the sample, and the threshold for determining ‘Low regularity’.’
➢ Comment 18: Row and Column Organization: Consider organizing rows and columns logically. For example, group similar outcome measures together (all PA outcomes, all SB outcomes). This can make the table flow better and help readers focus on one type of result at a time.
Response 18 – Thank you for your feedback. Table-1 was organised in 4 physical behaviour categories : PA amount, PA pattern, SB amount, SB pattern. We have now ensured that tables 2 and 3 also follow the same organisation for better flow and readability.
Graphs:
➢ Comment 19: Clarity and Labeling: Ensure that each graph is clearly labeled with axes titles, units of measurement, and a legend if necessary. For example, if a graph displays the changes in "PA Bout Length" across time, the y-axis should have "PA Bout Length (minutes)" clearly indicated. Include the exact p-values in the figure legend or on the graph itself (if space allows), as this will allow readers to interpret the statistical significance directly without referring back to the text.
Response 19 – All relevant graphs are already labelled with axes titles, units of measurement, and legends. However, we have also included the exact p-value in the figure legend for the significant group×time effect observed in figures 3 and 4 as requested.
➢ Comment 20: Simplification of Graphs: Some graphs may be difficult to interpret due to the number of data points or the way they are presented. Consider: Reducing the number of data points displayed if they are too dense.
Response 20 – Thank for your feedback. It was not possible to simplify graphs with dense data points like figures 1 & 2. This is due to the fact that we are showing individual data points for multiple participants across a seven-day monitoring period. This on the request of a previous reviewer.
➢ Comment 21: Using different colors or patterns to represent groups or time points, but make sure the colors are distinct and the legend is clear.
Response 21 – Thank you for your feedback. We will use distinct colours or patterns to differentiate groups or time points in the figures 3 and 4, ensuring the legend is clear and easy to follow. (We have used separate panels for different groups in figures 1 and 2.)
➢ Comment 22: Graphical Summary: Needs to be clear both visually and logically. You might consider adding a summary plot that includes both p-values and effect sizes for the most critical variables. A bar graph or line plot showing the changes across time for each group could be accompanied by a text box summarizing the key statistical results (p-values and effect sizes).
Response 22 – Thank you for your feedback. We respectfully disagree as the summary plot has to be distinct from the results we have shown in the paper. Our response is in line with the journal’s requirements for the summary to be a visually grabbing figure that summarises the entire article, not just a set of data.
Discussion:
➢ Comment 23: The conclusions drawn regarding the relationship between high regularity and intervention success are interesting but would benefit from further comparison with other studies in the field.
Are the results consistent with previous research, or do they contradict established findings? This comparison would give the results more context.
Response 23 – Thank you for your feedback. As noted in our previous response on sample size calculation, there is a lack of studies on physical behaviour regularity in older adults, making our findings novel. We have contextualized our results by aligning them with existing epidemiological and experimental evidence on physical behaviour and health. Additionally, we acknowledge the need for future studies to build on our findings by exploring psychological determinants of adherence, such as motivation, self-efficacy, and perceived barriers.
➢ Comment 24: While future studies are mentioned, more specific suggestions could be made regarding how interventions can target improving regularity or which interventions might be most successful for people with low baseline regularity.
Response 24 – We thank the reviewer for their feedback. Aside from the very specific suggestions we have outlined for future studies, we believe that more in-depth suggestions on how future studies may target improving regularity, or potential successful strategies (i.e. designing and justifying a potential future study’s methodology) is beyond the scope of this paper.
Areas for Improvement:
➢ Comment 25: Some minor grammatical issues and awkward phrasing should be revised for clarity and flow.
Response 25 - All grammatical issues and awkward phrasing have now been revised for clarity, thereby improving the flow in the revised manuscript.
➢ Comment 26: The discussion could benefit from further elaboration on how findings compare to existing literature.
Response 26 – Thank you for your feedback. This has already been addressed in a previous response. Please see the previous response to comment 23.
➢ Comment 27: Sample size issues, especially with the small control group, should be addressed by conducting a power analysis or explaining how this limitation affects the generalizability of the results.
Response 27 – Thank you for your feedback. This has already been addressed in a previous response. Please see the previous response to comment 8.
➢ Comment 28: Justifying statistical choices (chi-squared tests, sample size) and a more precise explanation of why these were the best options for this study will strengthen the credibility of the methods.
Response 28 – Thank you for your feedback. We have already provided a substantial written justification (>500 words) within the methodology section (section 2.7) justifying the selection of each statistical test. Nevertheless, the following text has now been added into the methods section:
‘For a Chi-squared test to be appropriate, the expected frequency count for each cell should generally be at least 5. We ensured this assumption was met by verifying that the number of participants in each classification category (e.g., high vs. low regularity) was sufficient for comparison. Specifically, all high-regularity groups had n=8, meaning comparisons were made against a frequency of n≥8.’
English Usage:
➢ Comment 29: Some minor grammatical issues should be revised. For example, in the phrase "proprotion of the study sample," there is a typo ("proprotion" should be "proportion"). Additionally, there are some occasional awkward sentence structures ("No baseline differences were found in the participants' PA patterns..."). Refining sentence flow and correcting these errors would improve readability.
Response 29 – Thank you for your feedback. All typos, such as "proprotion" (corrected to "proportion"), and awkward sentence structures have been revised for improved readability and precision.
➢ Comment 30: The phrasing in some areas could be more precise. For instance, terms like "average PA bout length" could be better contextualized for clarity. Define what constitutes an "average" PA bout or specify if it is across all participants or within specific groups.
Response 30 – Thank you for your feedback. Average PA bout length refers to the average length of a PA bout within a given monitoring day for a single participant. This was outlined in the original methodology and specified as being measured in minutes per day. Nevertheless, additional context has now been added where the term "average PA bout length" is discussed in depth within the discussion:
‘Average PA bout length refers to the average length of a PA bout within a given monitoring day for a single participant [XX].’
New references added below are:
Wullems, J.A., Verschueren, S.M., Degens, H., Morse, C.I. and Onambélé-Pearson, G.L., 2024. Concurrent validity of four activity monitors in older adults. Sensors, 24(3), p.895.
Line by line grammar and structure:
➢ Comment 31: L98-99
- "Recruitment packages (which included health questionnaires, participant information sheets,
and a pre-paid return envelope), were sent to all contacts (n=271) on a pre-existing research
database of older women."
o Correction: Remove the comma after "envelope."
o Revised: "Recruitment packages (which included health questionnaires, participant
information sheets, and a pre-paid return envelope) were sent to all contacts (n=271) on
a pre-existing research database of older women."
Response 31 – Thank you for your recommendation. The comma after "envelope" has been removed. Revised sentence: "Recruitment packages (which included health questionnaires, participant information sheets, and a pre-paid return envelope) were sent to all contacts (n=271) on a pre-existing research database of older women."
➢ Comment 32: L100-101
- "Inclusion criteria included being female, aged between 65-85y, and not knowingly allergic to
the physical behavior monitoring equipment."
o Correction: "Included" is used twice, so replace one of them for clarity.
o Revised: "Inclusion criteria were being female, aged between 65-85 years, and not
knowingly allergic to the physical behavior monitoring equipment."
Response 32 – Thank you for your recommendation. The repetition of "included" has been addressed. Revised sentence: "Inclusion criteria were being female, aged between 65-85 years, and not knowingly allergic to the physical behavior monitoring equipment."
➢ Comment 33: L110-111
- "During the familiarization visit participants filled various questionnaires (e.g. falls risk
assessment tool) and underwent anthropometric assessment (height and weight)."
o Correction: Add a comma after "tool" for clarity.
o Revised: "During the familiarization visit, participants filled out various questionnaires
(e.g., falls risk assessment tool) and underwent anthropometric assessment (height and
weight)."
Response 33 – Thank you for your recommendation. A comma will be added after "tool." Revised sentence: "During the familiarization visit, participants filled out various questionnaires (e.g., falls risk assessment tool), and underwent anthropometric assessment (height and weight)."
➢ Comment 34: L 120-121
- "Participants were given specific instructions depending on their allocated group."
o Correction: The sentence can be made more concise.
o Revised: "Participants received specific instructions based on their allocated group."
149-150
- "However, to the current authors' knowledge, PA and SB inter-day variability, have yet to be
investigated in the context of older adult populations."
o Correction: Remove the comma after "variability."
o Revised: "However, to the current authors' knowledge, PA and SB inter-day variability
have yet to be investigated in the context of older adult populations."
Response 34 – Thank you for your recommendation. The sentence has been revised for conciseness. Revised sentence: "Participants received specific instructions based on their allocated group." Comma also removed after variability.
➢ Comment 35: L 166-167
- "This led to our utilizing a monitoring period of sufficient duration to balance data quality and
participant comfort."
o Correction: "Our utilizing" sounds awkward.
o Revised: "This led us to use a monitoring period of sufficient duration to balance data
quality and participant comfort."
Response 35 – Thank you for your recommendation. The awkward phrasing "our utilizing" has been revised. Revised sentence: "This led us to use a monitoring period of sufficient duration to balance data quality and participant comfort."
➢ Comment 36: L 172-173
- "The .bin file was smoothed in the 'Data Convertor' Stage, into 10-second (s) epochs to optimise
the sensitivity to detect changes in physical behaviour."
o Correction: Remove the comma after "Stage" for proper flow.
o Revised: "The .bin file was smoothed in the 'Data Convertor' Stage into 10-second (s)
epochs to optimise the sensitivity to detect changes in physical behaviour."
Response 36 – Thank you for your recommendation. The comma after "Stage" has been removed. Revised sentence: "The .bin file was smoothed in the 'Data Convertor' Stage into 10-second (s) epochs to optimise the sensitivity to detect changes in physical behaviour."
➢ Comment 37: L 190-191
- "Protocols of 2.5-4.9 valid days hip accelerometery have previously been shown to be the
minimum number of valid monitoring days required for achieving a good intra-class correlation
co-efficient (ICC) of 0.80."
o Correction: "Accelerometery" should be "accelerometry," and the phrasing "have
previously been shown to be" could be more concise.
o Revised: "Protocols of 2.5-4.9 valid days of hip accelerometry have been shown to be
the minimum number of valid monitoring days required to achieve a good intra-class
correlation coefficient (ICC) of 0.80."
Response 37 – Thank you for your recommendation. "Accelerometery" has been corrected to "accelerometry," and the phrasing revised for conciseness. Revised sentence: "Protocols of 2.5-4.9 valid days of hip accelerometry have been shown to be the minimum number of valid monitoring days required to achieve a good intra-class correlation coefficient (ICC) of 0.80."
➢ Comment 38: L 304-305
- "The proportion of the study sample that therefore qualified as high regularity (i.e <25th
Percentile) is shown in Table 3."
o Correction: "Is" should be "was" for consistency in past tense.
o Revised: "The proportion of the study sample that therefore qualified as high regularity
(i.e., <25th Percentile) was shown in Table 3."
Response 38 – Thank you for your recommendation. We believe use of the word "is" is still appropriate in this case. “Was” implies that this metric was shown in table 3 but is no longer shown in table 3. Therefore we have kept “is” instead of “was”.
➢ Comment 39: L 346-347
- "Participants categorized as high regularity at baseline for average PA bout length (baseline interday variability: < 17%), exhibited a significantly greater increase over time (baseline: 15.8±3.6 mins, week 8: 20.8±7.5 mins, Δ%: 34±56%), when compared to low regularity participants."
o Correction: The phrase "exhibited a significantly greater increase" could be clearer.
o Revised: "Participants categorized as high regularity at baseline for average PA bout
length (baseline inter-day variability: < 17%) showed a significantly greater increase over
time (baseline: 15.8±3.6 mins, week 8: 20.8±7.5 mins, Δ%: 34±56%) compared to low
regularity participants."
Response 39 - The phrase "exhibited a significantly greater increase" has been revised for clarity. Revised sentence: "Participants categorized as high regularity at baseline for average PA bout length (baseline inter-day variability: < 17%) showed a significantly greater increase over time (baseline: 15.8±3.6 mins, week 8: 20.8±7.5 mins, Δ%: 34±56%) compared to low regularity participants."
➢ Comment 40: L 444-445
- "Future SB reduction studies should use greater sample sizes."
o Correction: "Greater" could be replaced with "larger" for clarity.
o Revised: "Future SB reduction studies should use larger sample sizes."
Response 40 – Thank you for your recommendation. "Greater" has been replaced with "larger." Revised sentence: "Future SB reduction studies should use larger sample sizes."
➢ Comment 41: L 484-485
- "Our study is further limited by the sole recruitment of older women, which limits the
generalizability of our findings."
o Correction: The sentence could be more concise.
o Revised: "Our study is limited by the exclusive recruitment of older women, which
affects the generalizability of our findings
Response 41 – Thank you for your recommendation. The sentence has been revised for conciseness. Revised sentence: "Our study is limited by the exclusive recruitment of older women, which affects the generalizability of our findings."

Reviewer 4 Report (New Reviewer)
Comments and Suggestions for Authors
The research is appropriate but certain Headings requires appropriate section number.
- Heading such as in section 2 "Experimental Design " "Physical behavior interventions" , "Physical Behaviour Assessment", "Physical Behaviour Classification", "Inter-Day Variability Assessment", "Threshold Determination ", "Statistical Analyses", .
- Table 3. A), B), C), Needs proper headings and heading numbers.
- Similarly Figure A to F needs proper headings.
- Save and close all track changes in the file.
- Compare your results with other benchmark methods proposed in your research method.
Author Response
- Summary
Thank you very much for taking the time to review this manuscript. Please find the detailed responses below and the corresponding revisions/corrections highlighted/in track changes in the re-submitted files.
- Point-by-point response to Comments
Comments 1: The research is appropriate but certain Headings requires appropriate section number. Heading such as in section 2 "Experimental Design " "Physical behavior interventions" , "Physical Behaviour Assessment", "Physical Behaviour Classification", "Inter-Day Variability Assessment", "Threshold Determination ", "Statistical Analyses".
Response 1: We agree and have, accordingly, now added numbers to the sections as per our reviewer’s recommendations.
Comments 2: Table 3. A), B), C), Needs proper headings and heading numbers.
Response 2: Agreed. We have, accordingly, added the headings in table 3.
Comments 3: Similarly Figure A to F needs proper headings.
Response 3: Agreed. We have, accordingly, expanded on the Legend of Figure 1 so that it is clearer what each figure represents. E.g. C-CON at Baseline etc
Comments 4: Save and close all track changes in the file.
Response 4: We will indeed upload a copy with tracked changes (as per the publishers’ requirements) as well as a CLEAN COPY (i.e. with no tracked changes)
Comments 5: Compare your results with other benchmark methods proposed in your research method.
Response 5: Thank you for your comment regarding the comparison of our results with other benchmark methods. However, we would like to clarify that the focus of our study was not on methodological validation but rather on the temporal variability of physical behaviour data. The methodological aspects, including the performance of the thigh-mounted triaxial accelerometer algorithms, have been addressed in our previous work (Wullems et al., 2017; Wullems et al., 2024). As our study primarily examines how the data varies over time, regardless of its computation, we do not feel it would be appropriate to introduce a dedicated discussion on methodological comparisons.
That said, we acknowledge the importance of clarifying our methodological approach and have added a brief statement in the first paragraph of the Discussion to reinforce this distinction.
This manuscript is a resubmission of an earlier submission. The following is a list of the peer review reports and author responses from that submission.
Round 1
Reviewer 1 Report
Comments and Suggestions for Authors
Thank you for your submission to the journal Sensors and contribution to the field.
In reading the paper, I wondered why the research was limited to just women? As described in the introduction differences have been reported present irrespective of gender. Could a sentence or two explaining this decision to the reader please be added.
The description of how to account for the Hawthorne Effect, why not just have them wear the accelerometer for 2 weeks, switching/charging it after week one, or perhaps even not even turning it on until the second week? This would create a more consistent timeframe of monitoring for all participants and better correct for transient changes in behavior. For that matter, tracking activity throughout the 8-weeks would have been the ideal. With a sample size of only 31, why not do that?
Regarding line 194 in the Materials and Methods… is this supposed to read SB in bouts of > 5 minutes duration? I see that it is written this way throughout the paper. What is the minimum duration required to consider something SB? If a person is sitting for 5 seconds, is that tallied in this count? How about 30 seconds? 1 minute?... The minimum threshold, if any, is impactful to the meaning of the results observed.
Additionally, would number the number of sustained bouts of SB be more impactful than the number of short bouts of activity? A decrease in short bouts of SB could be considered good or bad. If this number increased whilst total SB also decreased, that would be considered good. Yet, if it this value decreased whilst total SB time also decreased, that would also be good. It seems more meaningful and directionally consistent to look at the inverse, number of SB bouts above some minimal threshold, e.g. 5 minutes, in excess of which the SB is potentially deleterious to one’s health.
Line 196, y missing from ambulatory
I was confused by the equation on line 211: “(S(xi-xMean)2)/n)”.
I assume “S” represents the Greek sigma “S” and the 2 was supposed to appear as a superscript, such that this is just the equation for variance, or the standard deviation squared. If this is the case, the n-1 denominator would be the preferred equation, given this is a sample rather than a population. Moreover, reporting the standard deviation instead of the variance would make more sense, as this is the more commonly used metric, and Cochran’s theorem estimates hundreds of observations for calculating variance and tens for calculating SD. In any case, as coefficient of variation is derived from SD and SD is derived from variance, thus using coefficient of variation, variance, or SD does not resolve the issues listed in the preceding paragraph (number of valid days, sensitivity to outliers). To address these, it would be prudent to set min/max criteria, such as needing all 7 days of data. On that matter, using a set of 7 observations (7 days /week) to establish a coefficient of variation for each participant seems low, especially with an n of 31. In reference 20, the authors used a 30-day period to calculate coefficient of variation (30 observations per participant) for each participant, with an n of 26,935. The low number of observations and small sample size for data collection would seem to lack the power to support meaningful findings. And as for smaller means being more sensitive to change, this is not a problem, as a change of equal size could justifiably be viewed as relatively more impactful to a data set with a smaller mean. Incidentally, that is what the coefficient of variation is meant for, scaling the variance, not the other way around.
In relation to the above, it was unclear what values were used for the ANOVAs. Were raw values of the PA and SB metrics used, or were coefficients of variation for these metrics used for raw ANOVA values? The former would indicate meaningful changes in behavior, but not target the variance specifically. The latter, comparing coefficient of variation with ANOVAs, seems mathematically redundant, as coefficient of variation is equal to the root mean square error divided by the grand mean.
The results were reported clearly, and discussion grounded solidly in the results. However, as stated above, I do not believe there is sufficient data, in terms of the sample size, nor the duration of the pre/post observation periods to adequately support findings herein.
Reviewer 2 Report
Comments and Suggestions for Authors
The work presented here is of interest and aims to identify the variability of daily physical activity versus absolute or average values as a reference element in older women.
The study divides the women who met the inclusion criteria into three groups, two intervention groups (SBF and LIPA) and a control group (CON). It aims to make a comparison of the intra-group results between a baseline and an 8-week time point. In addition, it aims to compare the results between the groups. Lines 83-90 and 124-143
It also indicates that it compares women by criteria of high and low regularity in the intervention groups, in sedentary classification (ambulator & couch potato) and by their activity at baseline (active and inactive) lines 250-252.
The methodological process described fits into the analysis process, but in the result presentation no elements can be observed that could confirm the results obtained.
Table 1 presents the overall results of the 31 women without separating them into groups, it is not relevant for the study because it is intended to compare groups. It can serve as global data, but is not necessary for the analysis required. Table 2 provides the baseline data of the 31 women separated by groups, this table can provide the p-values showing the homogeneity, or not, of the sample. A table with the results from week 8 is needed to be able to see the results that occur after these weeks of intervention in the different groups, as well as to know if there are intergroup differences after the 8 weeks of intervention. In the work of Grant et al. 2020 the results at the two time points are shown in table 1, 2 and 3, indicating whether or not there were significant differences, although they lack the placement of the p-values as well as the effect size (Grant et al., 2020).
A differentiation is made between groups of active non-active, high and low variability and type of sedentary activity (lines 196-204 and lines 249-252). In the document it is not possible to identify these women, it is not known if this classification is intra-group (SBF, LIPA, CON) or is global of the 31 women, this conditions the comparisons and even the distribution of these women in the different groups, so that they are balanced groups and do not have a bias that conditions the intervention.
The day-to-day variation is presented in two graphs (fig 1 and 2) which are made with the 36 women, not the 31 women in the study, at baseline but not at week 8 post-intervention, nor is this variability shown in each group (SBF, LIPA, CON). With these data it is not possible to know how the intervention has influenced each group. A comparison of the variation in the high regularity and low regularity groups is made (fig 3 and 4) but this is only done at baseline and the n of each group is unknown. As in the previous comment it is not possible to know how the intervention has affected each group and whether or not it has changed the inter-group coefficient of variation.
The discussion is based on results that cannot be seen in the paper. Needs an update of all documentation in the paper to support the hypotheses and claims made.
Elements to note from the paper.
The abbreviation LIPA appears for the first time in the development of the text on line 69 should stand for the acronym "light intensity physical activity (LIPA)" and in the rest of the document use LIPA.
The units used in lines 188-195 are not correct, hrs.day-1 should be h·day-1 or h/day. In general, it happens in the dots and the -1, the dot is used as a decimal and should indicate multiplication and the -1 as a subtraction and is an exponent. It is well used in line 196 (≥8 h/day), it is advised to use the same style.
On line 205 the CV% equation is incorrect it should be (Standard deviation/average)·100 to be a percentage value.
The calculation of the individual coefficient of variation has to be modified the formula is wrong it should be (in attach file:
Grant, D., Tomlinson, D., Tsintzas, K., Kolic, P., & Onambele-Pearson, G. (2020). Displacing Sedentary Behaviour with Light Intensity Physical Activity Spontaneously Alters Habitual Macronutrient Intake and Enhances Dietary Quality in Older Females. Nutrients, 12(8), 2431. https://doi.org/10.3390/nu12082431
